# Ice Nucleation Activity of Alpine Bioaerosol Emitted in Vicinity of a Birch Forest

Teresa M. Seifried [1], Paul Bieber [1], Anna T. Kunert [2], David G. Schmale III [3], Karin Whitmore [4], Janine Fröhlich-Nowoisky [2] and Hinrich Grothe [1,*]

[1] Institute of Materials Chemistry, TU Wien, 1060 Vienna, Austria; teresa.seifried@tuwien.ac.at (T.M.S.); paul.bieber@tuwien.ac.at (P.B.)

[2] Multiphase Chemistry Department, Max Planck Institute for Chemistry, 55128 Mainz, Germany; anna.kunert@mpic.de (A.T.K.); j.frohlich@mpic.de (J.F.-N.)

[3] School of Plant and Environmental Sciences, Virginia Tech, Blacksburg, VA 24061-0390, USA; dschmale@vt.edu

[4] Service Center for Transmission Electron Microscopy, TU Wien, 1040 Vienna, Austria; karin.whitmore@tuwien.ac.at

[*] Correspondence: hinrich.grothe@tuwien.ac.at

**Abstract:** In alpine environments, many plants, bacteria, and fungi contain ice nuclei (IN) that control freezing events, providing survival benefits. Once airborne, IN could trigger ice nucleation in cloud droplets, influencing the radiation budget and the hydrological cycle. To estimate the atmospheric relevance of alpine IN, investigations near emission sources are inevitable. In this study, we collected 14 aerosol samples over three days in August 2019 at a single site in the Austrian Alps, close to a forest of silver birches, which are known to release IN from their surface. Samples were taken during and after rainfall, as possible trigger of aerosol emission by an impactor and impinger at the ground level. In addition, we collected aerosol samples above the canopy using a rotary wing drone. Samples were analyzed for ice nucleation activity, and bioaerosols were characterized based on morphology and auto-fluorescence using microscopic techniques. We found high concentrations of IN below the canopy, with a freezing behavior similar to birch extracts. Sampled particles showed auto-fluorescent characteristics and the morphology strongly suggested the presence of cellular material. Moreover, some particles appeared to be coated with an organic film. To our knowledge, this is the first investigation of aerosol emission sources in alpine vegetation with a focus on birches.

**Keywords:** bioaerosol; ice nucleation; alpine vegetation; birch; fluorescence microscopy; scanning electron microscopy

## 1. Introduction

Many organisms commonly found in the Alps, ranging from small pathogenic bacteria e.g., *Pseudomonas syringae* to large vascular plants, e.g., birch trees, are known to contain ice nuclei (IN) [1–3]. The distribution and transport of IN from alpine environment is still unclear. The main question tackled by the scientific community is whether IN from alpine vegetation especially with the presence of frost-resistant plants can be aerosolized and reach altitudes as high as the free troposphere. In the free troposphere, IN could trigger ice crystal formation of water droplets, which could lead to cloud glaciation, if respective altitudes are reached. In the absence of IN, cloud droplets would freeze homogeneously at temperatures below $-38\,°C$ [4]. If IN are present, however, the phase transition from liquid to solid water occurs at higher temperatures. The formed ice clouds do not only influence the Earth's radiation budget by interacting with solar radiation [5,6], but also play a key role in precipitation formation [7–9]. If precipitation reaches the land surface as rain, IN from biological surfaces can be aerosolized and transported further. Already in the 1960s, studies have shown that heavy rain events and storms are correlated with high

IN concentrations in the atmosphere [10,11]. More recently, field studies and laboratory measurements have shown that rain events over vegetation lead to a burst of biological IN [12–16]. Huffman, et al. [12] found that the burst of primary biological aerosol particles (PBAPs) above a forest with high IN concentrations is associated with rainfall. There are different theories and explanations about the aerosolization processes of these biogenic IN. First, the mechanical impact of rain droplets on wetted leaves could lead to the formation of airborne IN by ejecting ice nucleation active microorganisms or macromolecular IN [17]. Second, IN from biological surfaces can be washed off to the ground by precipitation and deposit on leaf litter, get incorporated into soil or transported further to rivers [18]. Strong winds and turbulences on the ground can subsequently lead to the aerosolization of soil particles carrying IN through abrasive dislodgment [19]. Further, Joung, et al. [20] suggested that biogenic particles in soil get aerosolized due to the mechanical impact of rain droplets releasing small bubbles. In the case of alpine vegetation, this process would lead to the formation and release of small droplets, some of which contain IN. In addition, microscopic vortices induced by raindrop impacts may also aerosolize ice-nucleating microorganisms, such as fungi [21]. The alpine region is rich with lakes and rivers. Ice-nucleating microbes from these aquatic environments can be aerosolized via different mechanisms such as bubble bursting [22,23] or fragmentation [24] possibly induced by strong winds [25]. For example, Powers, et al. [26] collected aerosols directly above a freshwater lake in Dublin, VA, USA, with a fixed-wing drone and detected culturable microorganisms, some of which were ice nucleation active. Furthermore, relative humidity triggers the biological activity of various species [27,28], and sporulation processes of various fungi are enhanced at high relative humidity [12,29,30]. Indeed, some fungal species are among the most active IN; *Fusarium acuminatum* can initiate freezing at around $-5\,°C$ [31,32]. Furthermore, pollen grains, some of which are known to contain IN, e.g., *Betula pendula* [33,34], are released during the spring season and start to germinate or rupture during wet conditions due to the enhancement of osmotic pressure [35]. This leads to the release of sub pollen particles with aerodynamic diameters of less than 3.3 μm [36,37]. These small particles have the potential to reach higher altitudes compared to intact pollen grains. Thus, rainfall and increased relative humidity are important for the release of primary and secondary biological aerosol particles.

In a recent study, we discovered that ice nucleating macromolecules (INMs) are extracted from the surface of birches by rain [38]. Here, we want to test the hypothesis if IN from alpine environments are emitted into the lower atmosphere near major vegetation sources (with a focus on silver birches). To find first evidence for this hypothesis, we investigated the abundance of IN in an alpine environment in the Austrian Alps. Aerosol samples were collected with an impinger both stationary and in air using a rotary wing drone. For the aim of microscopic characterization of aerosol particles below the canopy, we used a cascade impactor as an additional sampling tool. Observed IN concentrations were examined in the context of meteorological parameters including temperature, relative humidity, and rainfall. This paradigmatic work extends the aerosol-sampling technology highlighted by Bieber, et al. [39] by (1) measuring ice nucleation activity (INA) at lower limit of quantification with the high-throughput Twin-plate Ice Nucleation Assay (TINA), (2) focusing on IN concentrations on the ground and above the tree canopy of birches and, (3) extending fluorescence and electron microscopy from detection to assignment of biopartides. The specific objectives of our work were to: (i) investigate the ice-nucleation activity of aerosols sampled at different altitudes in an alpine environment near major sources of vegetation, (ii) examine potential associations of IN concentrations in alpine environments with meteorological parameters, and (iii) characterize biological aerosol particles according to their morphology using fluorescence and scanning electron microscopy (SEM).

## 2. Experiments

### 2.1. Field Campaign

The campaign took place on three days in August 2019 in the Alps of Upper Austria (UA) (GPS: 47.50052, 13.54740). The sampling site was well chosen for this purpose and is situated at the end of a deep, narrow valley where a lake (Hinterer Gosausee) is only surrounded by alpine vegetation and high mountains (see Figure 1a). Thus, this remote situation of the site gives the opportunity to investigate biological aerosols in an area with no direct anthropogenic emission sources, but also long-distance transport, e.g., from industrial areas or from the Sahara is thought to play a minor role. The sample collection took place in the direct environment of a birch forest (*Betula pendula*). We chose rather foggy and wet days for sampling, since high relative humidity activates various biological processes, leading to a burst of bioaerosols. To sample the aerosol, we used the system DAPSI (Drone-based Aerosol Particles Sampling Impinger/Impactor), which was described previously by Bieber, et al. [39]. In short, DAPSI includes a self-build impinger, a cascade impactor, and sensors to measure environmental parameters (e.g., temperature and relative humidity), which are attachable to DJI Phantom 4 rotary wing drones. The impactor has four filter stages with the following cut-offs: stage A: 2.5 μm, stage B: 1.0 μm, stage C: 0.5 μm, and stage D: 0.25 μm. We used pre-baked aluminum foils (460 °C for 5 h) as impactor substrates. During sample collection, the flow rate was set to 9 L min$^{-1}$. The sterile impinger vial was filled with 15 mL Milli-Q$^{®}$ water prior to sample collection and operated with 1 L min$^{-1}$. The collected impinger samples were cooled for transport and stored at −20 °C at the laboratory. Milli-Q$^{®}$ water was of Millipore grade (specific resistivity ≥ 17.5 MΩcm$^{-1}$, 25 °C), obtained from the Simplicity$^{®}$ UV water purification system (Merck, Darmstadt, Germany), and filtered with a 0.22 μm filter (Millipak express 20, Merck, Darmstadt, Germany). Further, it served as a control sample in ice nucleation measurements of impinger samples.

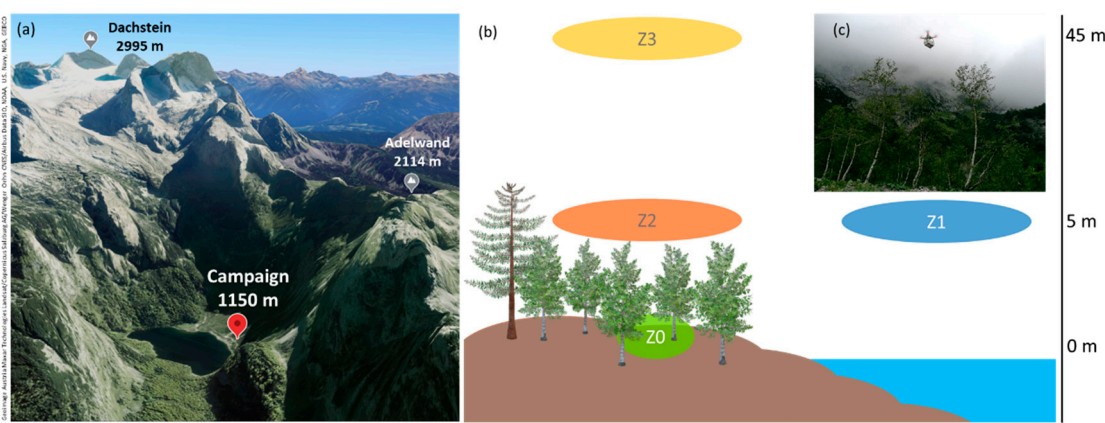

**Figure 1.** (**a**) A picture of the sampling location. The spot of the campaign is highlighted in red. The image was adapted from Google Earth ©, https://earth.google.com/ (accessed on 25 February 2021). (**b**) Zones in which aerosol collection took place related to altitude scale (right side). Zone 0 (Z0—green) is directly under the trees at 1 m altitude, zone 1 (Z1—blue) is above the lake (~5 m altitude), zone 2 (Z2—orange) is directly above the canopy of the birches (~5 m altitude), and zone 3 (Z3—yellow) is at about 45 m above the forest. (**c**) A picture of the sample collection with the drone in zone 2.

Air samples were taken both stationary on ground and in line with the previous study [39] with the rotary wing drone in different zones (see Figure 1b,c). Zone 0 is located directly under the trees and sampling was only carried out stationary at 1 m altitude. There, we set up a roofing made out of a plastic foil (about 1 × 1 m), approximately 30 cm above the sampling inlet to avoid rain droplets interfering with the measurement devices. Furthermore, we defined zone 1 to be the area above the lake, serving to collect reference samples at an altitude of approximately 5 m. The flights in zone 1 were about 130 m away from the shore of the lake. Zone 2 is directly above the birches' canopy (~5 m altitude) and

zone 3 is approximately 45 m above the ground level of the birch forest. In contrast to the campaign taking place in June 2019 [39], we focused on investigating the bioaerosol burst in the middle of vegetation with the impactor setup (zone 0). Stationary (zone 0) and drone samples (zone 1, 2, and 3) were collected with the impinger system. The samples were named after the location (UA = Upper Austria) and numbered from 3 to 5 for each day (UA3, UA4, and UA5), since UA1 and UA2 correspond to the first study in June 2019 [39]. On the first day of the field campaign in August (UA3), only stationary samples were collected, since the rainfall was too strong to fly with the drone. An overview of the field campaign including information on the samples, dates, and sampling parameters is given in Table 1.

**Table 1.** Overview of the field campaign including the sample ID, date, the zone, and altitude in which samples were collected, the starting time of the corresponding sampling collection, as well as the flow rate and the resulting air volume. Impinger is abbreviated as "Imp" and impactor as "Pac". Flow calibration was conducted every day prior to the measurements.

| Sample ID (Day-Device-Zone) | Date | Zone | Altitude (m) | Start Time (CEST) | Duration (min) | Flow Rate (L min$^{-1}$) | Air Volume (L) |
|---|---|---|---|---|---|---|---|
| UA3-Imp-Z0 | 21. 08. 2019 | 0 | 1 | 18:50 | 10 | $1.00 \pm 0.03$ | 10.0 |
| UA3-Pac-Z0 | 21. 08. 2019 | 0 | 1 | 18:50 | 10 | $8.92 \pm 0.04$ | 89.2 |
| UA3-Imp-Z0 | 21. 08. 2019 | 0 | 1 | 19:23 | 50 | $1.00 \pm 0.03$ | 50.0 |
| UA3-Pac-Z0 | 21. 08. 2019 | 0 | 1 | 19:23 | 50 | $8.92 \pm 0.04$ | 446 |
| UA4-Imp-Z0 | 22. 08. 2019 | 0 | 1 | 20:01 | 50 | $1.07 \pm 0.04$ | 53.5 |
| UA4-Pac-Z0 | 22. 08. 2019 | 0 | 1 | 20:01 | 50 | $8.94 \pm 0.06$ | 447 |
| UA4-Imp-Z1 | 22. 08. 2019 | 1 | 5 | 18:48 | 10 | $1.07 \pm 0.04$ | 10.7 |
| UA4-Imp-Z2 | 22. 08. 2019 | 2 | 5 | 18:28 | 10 | $1.07 \pm 0.04$ | 10.7 |
| UA4-Imp-Z3 | 22. 08. 2019 | 3 | 45 | 19:25 | 10 | $1.07 \pm 0.04$ | 10.7 |
| UA5-Imp-Z0 | 23. 08. 2019 | 0 | 1 | 19:54 | 10 | $1.08 \pm 0.02$ | 10.8 |
| UA5-Pac-Z0 | 23. 08. 2019 | 0 | 1 | 19:15 | 50 | $9.08 \pm 0.06$ | 454 |
| UA5-Imp-Z1 | 23. 08. 2019 | 1 | 5 | 18:55 | 10 | $1.08 \pm 0.02$ | 10.8 |
| UA5-Imp-Z2 | 23. 08. 2019 | 2 | 5 | 19:10 | 10 | $1.08 \pm 0.02$ | 10.8 |
| UA5-Imp-Z3 | 23. 08. 2019 | 3 | 45 | 19:25 | 10 | $1.08 \pm 0.02$ | 10.8 |

Aerosol samples obtained from the impinging system were analyzed for INA using the high-throughput droplet freezing assay TINA [40]. Furthermore, we analyzed the impactor samples using fluorescence microscopy and SEM.

### 2.2. Ice Nucleation

The aerosol samples collected from the impinging system were thawed shortly before the IN measurements. Each sample was diluted 1:10 and 1:100 with pure water to obtain the full ice nucleation spectrum. Of each dilution, 96 droplets (3 μL) were tested in TINA with a continuous cooling rate of 1 °C min$^{-1}$ from 0 to −30 °C with a temperature uncertainty of ±0.2 °C [40]. Prior to the calculation of the cumulative number of IN, $n_{IN}(T)$, we subtracted the background respectively to distinguish between freezing events from the sample and background water. First, ice nucleation data were binned into 0.2 °C intervals. Further, we calculated the differential nucleus concentration $k(T)$ of background water and samples using the following equation [41–43]:

$$k(T) = -\frac{1}{V \cdot \Delta T} \cdot ln\left(1 - \frac{\Delta N}{N(T)}\right) \tag{1}$$

where $V$ is the droplet volume, $\Delta T$ the temperature interval (0.2 °C in our experiments), $\Delta N$ the number of droplets that freeze during $\Delta T$, and $N(T)$ the number of unfrozen droplets at the beginning of a temperature bin.

To obtain the corrected $k(T)$ and hence, account for the background contribution in the samples, we performed the following:

$$k_{corr}(T) = k_{sample}(T) - k_{BG}(T) \qquad (2)$$

where $k_{sample}(T)$ is the differential nucleus concentration of the sample and $k_{BG}(T)$ of the background. Please note that in case of sample treatment (filtration, heat) a water blank, which was treated in the same manner, was used for background correction. By inverting Equation (1), the corrected fraction of frozen droplets (*FF*) can be obtained as shown in David, et al. [43]. The number of IN per volume can then be calculated using the Vali formula [41]:

$$n_{IN}(T) = -\frac{ln(1 - FF) \cdot d}{V} \cdot c \qquad (3)$$

where $d$ is the dilution factor, $V$ the droplet volume, *FF* the fraction of frozen droplets and $c$ the impinger sample volume divided by the sampled air volume.

To test if the ice nucleation assay works properly, pure water was measured at the beginning of each measurement day. Pure water was obtained by Barnstead™ GenPure™ xCAD Plus water purification system (Thermo Scientific, Braunschweig, Germany). The water was autoclaved at 121 °C for 20 min, filtered through a sterile 0.1 µm pore diameter polyethersulfone (PES) vacuum filter unit (VWR International, Radnor, PA, USA), and autoclaved again [40]. Aliquots of a selected aerosol sample (UA4-Imp-Z0) were filtered successively through a 0.1 µm PES syringe filter (Acrodisc®, Sigma-Aldrich, Taufkirchen, Germany) as well as a 300,000 and a 100,000 Da molecular weight cut-off PES ultrafiltration unit (Vivaspin®, Sartorius AG, Göttingen, Germany). The IN concentration was measured after each filtration step. Another aliquot of the selected aerosol sample (UA4-Imp-Z0) was incubated at 98 °C for 1 h to investigate the thermal stability and thus, to be able to draw conclusions about the biological fraction of the IN in line with previous studies [18,44]. The ice nucleation measurement uncertainties were calculated according to Kunert, et al. [40] and are visualized as error bars in the corresponding graphs. A comparison of the freezing curves from pure water to Milli-Q® water is given in the Appendix A, Figure A7.

### 2.3. Microscopy (Fluorescence, SEM)

Fluorescence microscopy pictures were recorded on a Nikon Eclipse Ci-L microscope (Nikon, Japan). The instrument is equipped with a 100 W metal-halide light source (Eclipse Ci-L, Nikon, Japan), a high-definition c-mount camera (DS-Fl1c, Nikon, Japan), and a 1/1.8 inch color CMOS image sensor. The microscope contains an epifluorescence unit including an excitation filter at 465–495 nm, a dichroic mirror at 505 nm, and an emission filter at 515–555 nm. The observation of our samples was conducted with two different objectives (10× and 40× magnification). The settings including light exposure time and analogue gain was modified respectively.

Impactor foils were further investigated under a scanning electron microscope of the type FEI Quanta™ 200 FEGSEM. A section of the sample foil was cut out and sputtered with a layer of Au/Pd. Afterwards, the prepared sample was transferred into the instrument and investigated.

### 2.4. Aerosol Particle Characterization

Aerosol samples collected parallel to impinger samples with highest IN concentration (UA4-Imp-Z0) were analyzed in more detail according to their morphology. The size distribution of particles sampled on the impactor stages A and D was determined by counting the particles on the impactor foils in well-defined areas in microscopic images using the program ImageJ. Three areas (200 µm × 200 µm) were selected randomly on fluorescence and SEM images of stage A. In case of stage D, we selected one single high-resolution SEM image with an area of 10.2 × 6.8 µm (for stage D, we did not take fluorescence images into account due to the limited resolution). By using the software ImageJ, particles were marked with pixels manually and diameters were calculated by referring to the surface (see

Appendix A, Figure A5). To calculate the concentration, we multiplied the corresponding selected area with the overall sampling area and divided it by the sampled air volume. The number concentration was normalized to the bin size and a log-normal distribution was fitted.

Furthermore, aerosols were classified into non-fluorescent and auto-fluorescent aerosol particles (FAP), and FAP were further divided regarding their shapes. In more detail, we took two images from the same spot on the impactor foil, one obtained from fluorescence microscopy and the other from scanning electron microscopy. We removed the black background of the fluorescence image and stacked it with the corresponding SEM picture (see Appendix A, Figure A6). Data evaluation was performed manually using the software ImageJ. In total, 302 particles were counted and attributed to a specific classification (global, ornamented, elongate, and cylindric forms) in the 216 × 278 µm selected area.

## 3. Results

### 3.1. Meteorological and Environmental Data

Throughout the campaign, it rained at least once a day. On UA3, we took samples during rainfall. Since precipitation continued that day, we were not able to fly with the drone, and thus, we only measured stationary samples from the ground (zone 0). On the remaining days (UA4, UA5), the measurements were performed after rain events, under foggy conditions, both on ground and in air with the drone. In Figure 2, weather data (relative humidity, temperature, and precipitation) adapted from simulated weather data (www.meteoblue.com, accessed on 30 October 2020) were compared to sensor data recorded with the onboard setup. The sensor data recorded during the campaign show that the first day of the field measurements (UA3) was the coldest with ambient temperatures between 12.7 and 13.9 °C. The following day (UA4) was warmer with temperatures between 15.0 and 15.9 °C. On the warmest day (UA5), temperatures reached 18.0 °C in zone 3 and 18.2 °C in zone 0. The temperature trend of the simulated data is in line with the measured values (maximum deviation was not higher than 1.7 °C). The relative humidity (RH) was above 79% every day. According to the sensor values, it was highest in zone 0 on UA4 reaching 91%. According to meteoblue®, however, the relative humidity was highest on UA3.

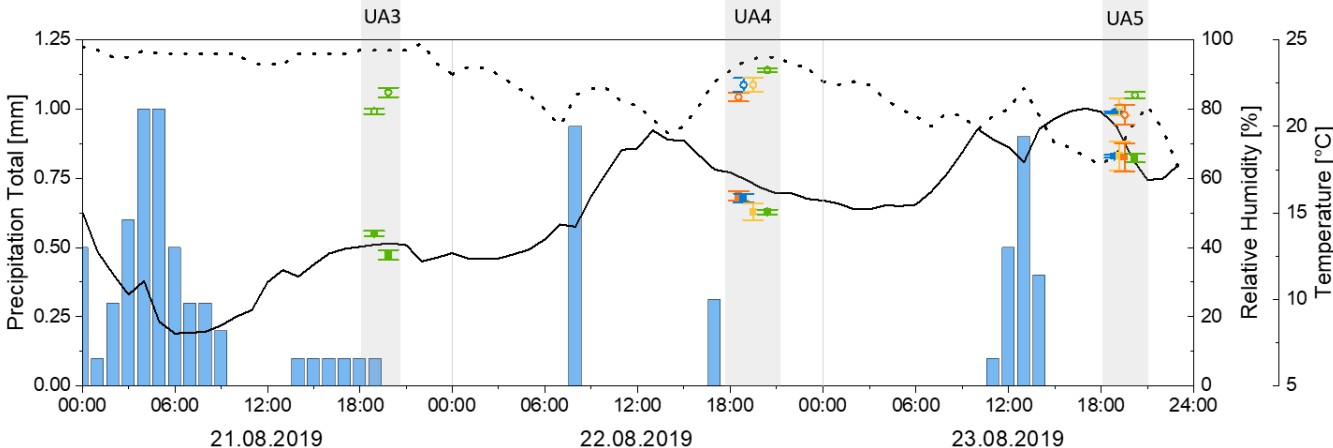

**Figure 2.** Simulated environmental data (www.meteoblue.com, accessed on 30 October 2020) of precipitation (blue bars), relative humidity (dashed line), and temperature (solid line) on the three sampling days compared to onboard sensor data recorded during field measurements (UA3, UA4, UA5): relative humidity (hollow circles) and temperature (filled squares); green represents zone 0, blue zone 1, orange zone 2, and yellow zone 3.

### 3.2. Ice Nucleation

The ice nucleation data of the collected impinger samples revealed that IN were present in the vicinity of the birch forest. No ice nucleation higher than the blank level

(Milli-Q® water; see section Experiments) was detected above the lake (zone 1). Figure 3 shows the corresponding freezing curves and the cumulative number of IN as a function of temperature. On the first day, the sample with a lower air sampling volume (UA3-Imp-Z0$_{10}$, 10 L air) did freeze slightly above blank level (Figure 3a) with a small shoulder at around −25 °C. The second sample of that day (UA3-Imp-Z0$_{50}$, 50 L air), however, started to freeze at about −13 °C. The freezing curve shows two intervals (−13 to −24 °C and −24 to −30 °C) in which the course of the curve increases gradually. The cumulative spectrum of UA3-Imp-Z0$_{50}$ shows a gradual increase beginning at −13 °C towards −30 °C. Between −23 and −29 °C, the cumulative spectra of zone 0$_{10}$ is rather similar to zone 0$_{50}$ (Figure 3b). Furthermore, the sample with the highest observed IN activity and the highest IN concentrations was collected on the second day (UA4-Imp-Z0) (Figure 3c,d), directly after rainfall. The freezing curve for this sample was rather steep at about −20.0 °C. The concentration of IN rose between −13.6 and −24 °C from $1.0 \times 10^0$ to $2.3 \times 10^3$ L$^{-1}$. Moreover, IN were detected in samples collected with the rotary wing drone in zone 2 and 3. It needs to be taken into consideration when comparing the freezing curves of the different zones in UA4, that the sampling periods differed (zone 0: 50 L air/sample, zone 2 and 3: 10 L air/sample). The freezing curves of both zone 2 and 3 have the same shape between −21 and −25 °C. At −25 °C, however, zone 3 begins to increase more steeply. The cumulative spectra reveal more clearly that zone 2 started to freeze at higher temperatures. On the last day of the field campaign (UA5), IN were present in zone 0 with a concentration of $2.5 \times 10^1$ to $7.7 \times 10^2$ L$^{-1}$ from −20 °C to −24 °C. Furthermore, IN were found to be present in zone 3. However, no significant amount of IN were detected above blank level in zone 2 (Figure 3e,f).

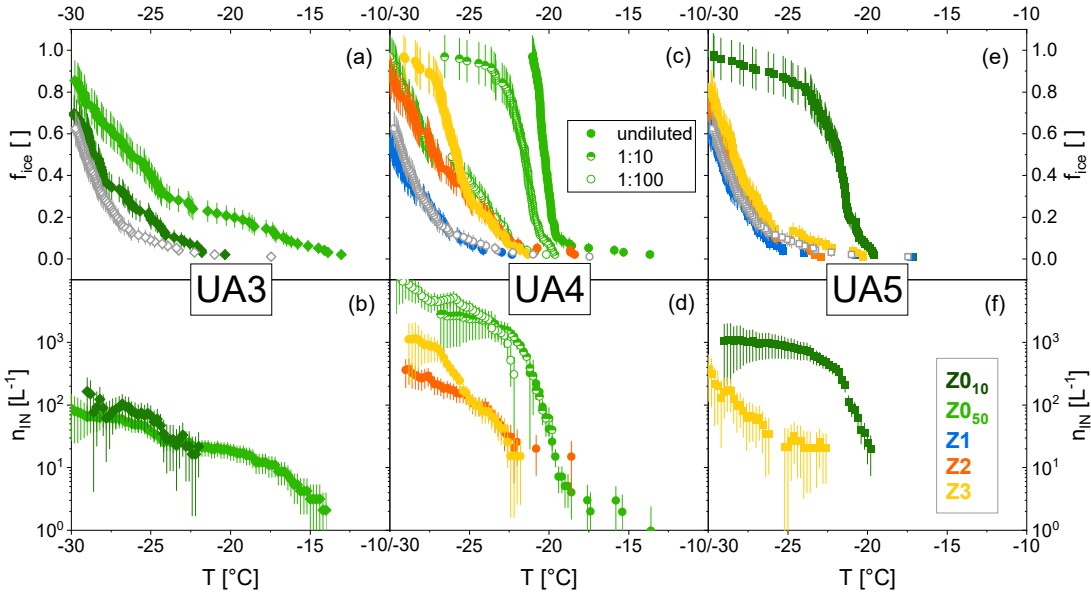

**Figure 3.** (**a**,**c**,**e**) Original freezing curves and (**b**,**d**,**f**) corresponding, background corrected cumulative number of ice nuclei ($n_{IN}(T)$) of samples taken at different altitudes (see Figure 1) in the direct environment of birches on the three sampling days. Zone 0 (Z0) is green, zone 1 (Z1) blue, zone 2 (Z2) is orange, and zone 3 (Z3) is yellow. Sample UA4-Imp-Z0 was diluted to obtain the cumulative ice nuclei spectrum until −30 °C (half−filled symbols correspond to a dilution factor of 10 and empty circles of 100). Grey circles in the freezing curves indicate the blank samples (Milli-Q® water).

To further characterize the collected IN, filtration and heat treatment experiments were performed on the sample with the greatest INA (UA4-Imp-Z0). In Figure 4, the cumulative number of IN ($n_{IN}(T)$) of the pre-treated samples are plotted as a function of the temperature. After filtration through a 0.1 μm syringe filter, no INA between −15.0 and −19.0 °C was measurable. IN concentrations above −19.0 °C, however, remained similar to the untreated sample. Although the filtrate obtained upon 300 kDa filtration lost its ice

nucleation between −15 °C and −20.0 °C, 5% of the original cumulative number of IN was found at −24 °C. In addition, the freezing curve of the 100 kDa filtrate was slightly above the level of the blank sample (see Appendix A, Figure A1). Hence, INA of airborne IN diminished after 100 kDa filtration. In addition, heat treatment led to a strong decrease in IN concentration within the whole spectrum (see Figure 4, red circles). At approximately −24 °C, however, the cumulative number of IN of the heat-treated sample increased.

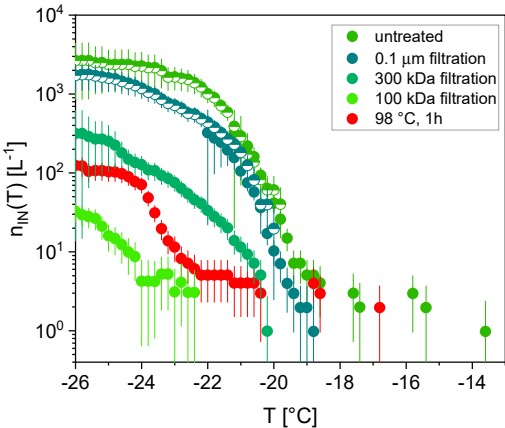

**Figure 4.** Cumulative number of ice nuclei ($n_{IN}(T)$) spectra of UA4-Imp-Z0; untreated (filled circles, green), filtrated with 0.1 μm cut-off (hollow circles, turquoise blue), 300 kDa cut−off (hollow circle, emerald green), 100 kDa cut-off (hollow circles, light green) and heated, 98 °C for 1 h (hollow circles, red). Half-filled symbols correspond to a dilution factor of 10.

### 3.3. Microscopy (Fluorescence, SEM)

Microscopic images reveal a deeper insight into the aerosol composition sampled in this study. On UA4, where impinger samples showed highest INA, a large number of FAP was observed (UA4-Pac-Z0). Impactor foils investigated with fluorescence microscopy (see Figure 5) reveal that most of the FAP did impact on stage A (cut-off at 2.5 μm) and stage D (cut-off at 0.25 μm). In comparison, a significant lower amount of FAP were observed in stage B and C (see Appendix A, Figure A3). In contrast, on day UA3, only a few considerably large particles (average diameter about 6 μm) were on stage A and a few small particles on stage D (see Appendix A, Figure A2). On UA5, noticeably the same amount of FAP were observed on stage A and D compared to UA4. Additionally, a small number of FAP were visible on stage B (see Appendix A, Figure A4). At a higher magnification (Figure 5b), FAP on stage A show defined cellular shapes (cylindric, longitudinal, and global) indicating biological FAP to be the main component. On stage D, a very heterogeneous mixture (agglomerates and unshaped particles) became visible (Figure 5d). However, FAP on stage D could not be resolved entirely with fluorescence microscopy because the size and shape were below the Abbe diffraction limit.

SEM images of the sample with the greatest INA (UA4-Pac-Z0) revealed detailed information about the morphology of the particles (including non-fluorescent particles). In stage A, a variety of particles were observed (Figure 6a). Mostly cylindric, longitudinal, and global particles in line with observations from fluorescence microscopy were visible. These observed shapes and sizes indicate potential spores of fungi and bacteria (see Figure 6b–d). In particular, several roundish spores with smooth surfaces and ornamented spores were observed. Some particles, however, had rather complex morphologies and irregular shapes (see Figure 6d). Many observed particles appeared to be coated with a film, or are collapsed soft particles with a hard core, as shown in Figure 6e. Furthermore, the SEM images of stage D (Figure 6f) revealed a high number of accumulation mode particles. The morphology of particles is rather undefined, and agglomerates are observed.

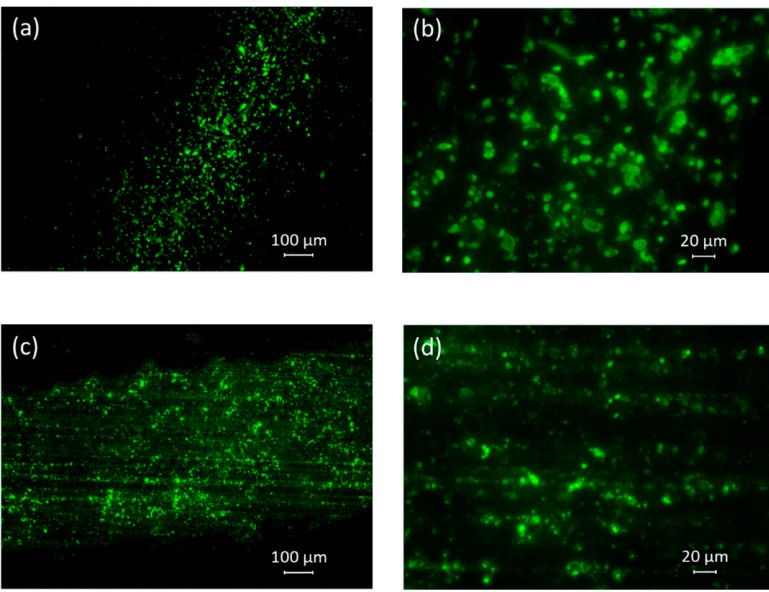

**Figure 5.** Fluorescence microscopy images of impactor foils sampled on UA4-Pac-Z0. (**a**) stage A, 10×, (**b**) stage A, 40×, (**c**) stage D, 10×, and (**d**) stage D, 40×.

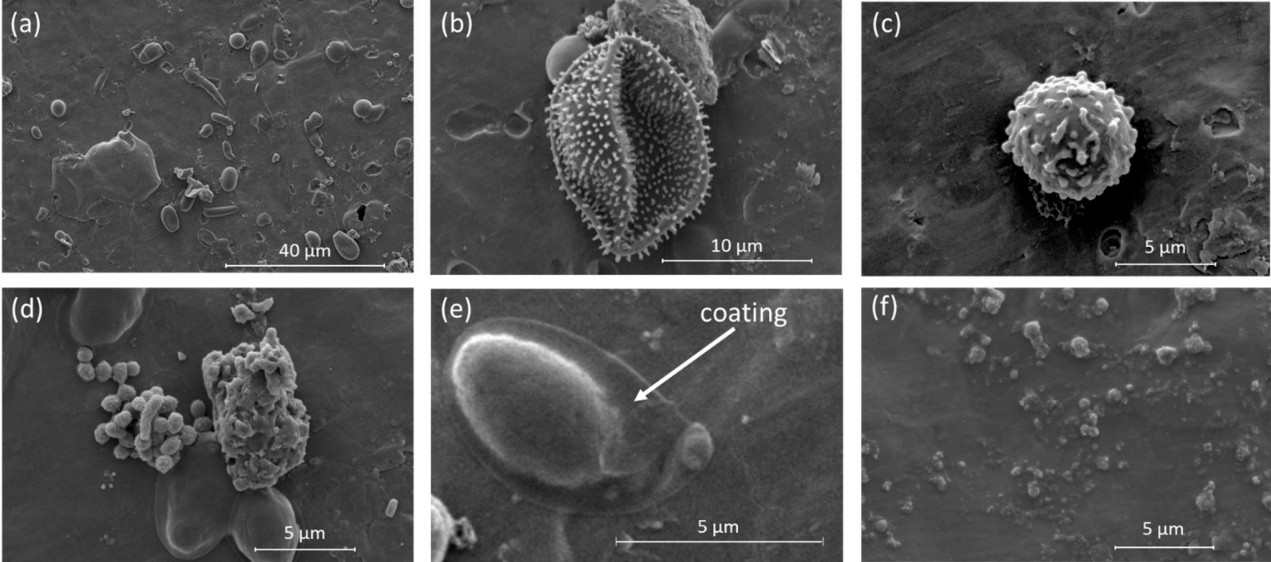

**Figure 6.** SEM images of impactor samples from UA4-Pac-Z0. (**a**) Overview of impacted particles on stage A; (**b**,**c**) fungal spores; (**d**) an agglomerate of bacteria and a non-biological particle; (**e**) a coated particle; (**f**) overview of impacted particles on stage D.

## 3.4. Aerosol Particle Characterization

The histogram of particles on stage D shows a log-normal distribution, with the mean particle diameters between 0.1 μm and 0.2 μm (Figure 7a). Thus, particles on stage D are classified to accumulation mode. Moreover, also the histogram of stage A reveals a log-normal distribution of particles (Figure 7b). The mean particle diameter is between 3 and 5 μm, assigning the particles to coarse mode. The size distribution determined with fluorescence images is shifted to larger diameters, since strong fluorescence light led to a blurring effect of the particles' interfaces, resulting in an overestimation of particle sizes. Furthermore, a fitted log normal distribution of both aerosol modes (Figure 7c) indicates that the mean diameter of stage D particles is 0.17 μm and of stage A 4.0 μm using SEM and 5.4 μm using fluorescence images for evaluations. The number concentration of the

accumulation mode on stage D is three orders of magnitudes higher than the coarse mode aerosol on stage A.

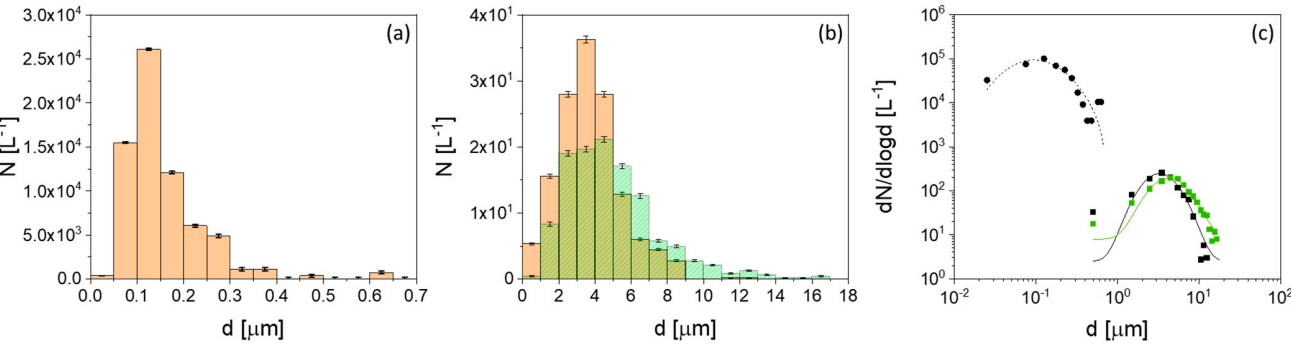

**Figure 7.** Number and particle size distribution of (**a**) accumulation mode particles on stage D and (**b**) coarse mode on stage A (evaluation with SEM (orange) and fluorescence microscopy (green) of UA4-Pac-Z0. (**c**) Log normal particle distribution of both modes.

Single particle analysis of the coarse mode provides insights into qualitative information of the sampled particles (see Figure 8). The stacked microscopic images obtained from SEM and fluorescence microscopy show that 80% of particles in the respective area are auto-fluorescent (see Figure 8a). Further analysis of the FAP focusing on the shapes of particles reveal that most particles are cylindrical. Additionally, 10% were counted as global, 4% ornamented, and 5% elongate. In addition, one major part (34%) of the counted particles could not be classified due to irregular shapes or agglomerates of small particles. Furthermore, some particles were coated possibly with an organic film, as seen in Figure 6e. However, in this analysis coating was not always clearly visible and therefore not subdivided into a class.

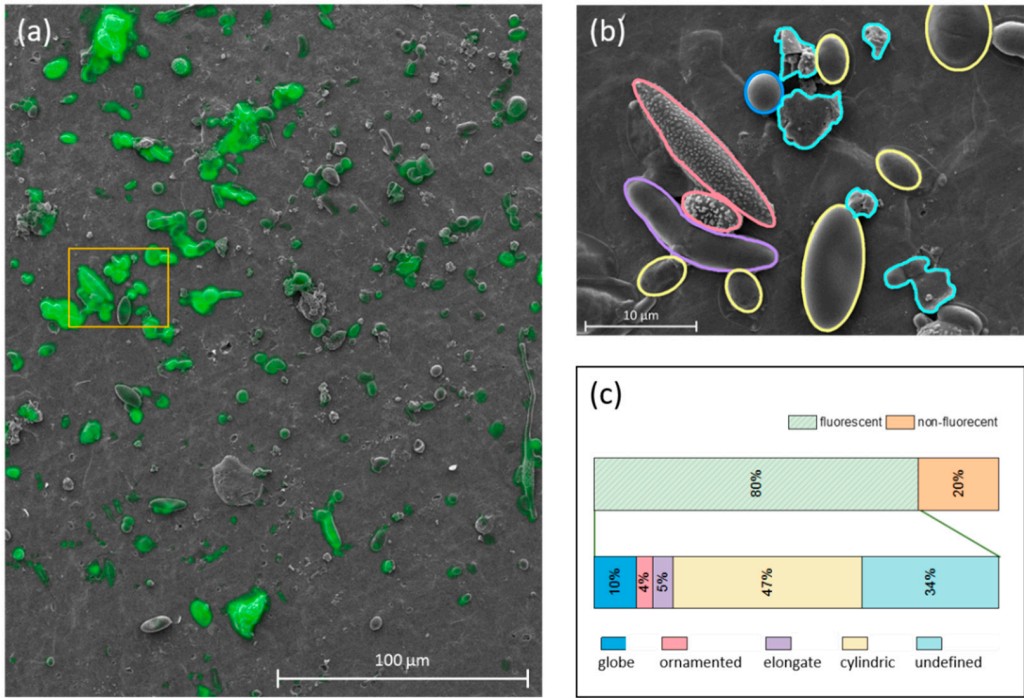

**Figure 8.** (**a**) Stacked microscopic fluorescence and SEM images differentiate FAP from non-fluorescent particles. (**b**) Shape classification of fluorescent particles into global (blue), ornamented (red), elongate (violet), and cylindric (yellow) forms and (**c**) results of the classification.

## 4. Discussion

Organisms living in alpine environments, characterized by low temperatures and short frost-free periods, need to cope with harsh conditions. Plants in these alpine environments have developed mechanisms to survive long winters and freeze/thaw cycles [45] even occurring during summer times [46]. Birches, for example, which grow not only in the Alps but also in the boreal forest, are known to be frost-hardy. Burke, et al. [47] proclaims that birch trees are able to survive experimental freezing down to −196 °C. In general, plant cells are at risk to get lethally damaged by ice at sub-zero temperatures, which is why some plants induce controlled freezing in their extracellular spaces through the release of IN, leading to freezing at higher temperatures [47–49]. Furthermore, these IN cause ice crystals to form in extracellular space by withdrawing water from the adjacent cells. The removal of liquid water from the intine of these cells leads to a higher supercooling capacity ensuring survival. Hence, IN are essential for the regulation of freezing in some plants. An open question is if these IN can get airborne and how far these IN can be transported. In principle, IN could trigger ice nucleation in cloud droplets if respective altitudes are reached, influencing the Earth's radiation budget as well as the hydrological cycle. Even though biological IN trigger freezing at highest temperatures, a clear evidence and source appointment during atmospheric IN measurements is lacking. Furthermore, the transport pathways of biogenic particles from the land surface up to mid altitudes are rarely elucidated. Thus, local and regional studies are inherently necessary. In this exemplification study, we collected aerosol samples in the direct environment of a birch forest over three days in August 2019 at a single site in the Austrian Alps, both stationary on ground and in air using a rotary wing drone. Samples were taken with an impinger and a cascade impactor.

On the first day of the study (UA3), collected samples were ice nucleation active, but with rather low concentrations. Since it was raining during the whole sampling period on that day, we assume the low concentrations to be evoked by wet deposition, which can occur due to falling rain droplets. On the next day (UA4), the highest IN concentrations ($n_{IN}$ (−24 °C) = $2.3 \times 10^3$ L$^{-1}$) were detected near the land surface (zone 0) after rain. In contrast to UA3, sampling was performed after the rain event at rather foggy conditions where wet deposition is assumed to play a minor role. In literature, different studies have observed that rain events over land surfaces lead to a burst of PBAPs [50,51] and biogenic IN [12,13,15,16]. In particular, Rathnayake, et al. [50] describes that not only rainfall but rather the interplay between rain and temperature determines which particles are emitted from the terrestrial biosphere. Furthermore, we found IN above the birches' canopies (zone 2) and up to 45 m altitude (zone 3) on the second sampling day. The sampling site is at 1150 m a.s.l. and surrounded by high mountains (e.g., Adelwand 2114 m a.s.l., Dachstein 2995 m a.s.l.), and thus, aerosol can be transported vertically into the free troposphere by thermal convection [52]. However, the concentration of the IN and the freezing activity are significantly lower compared to the ground samples and similar to diluted samples. This indicates that IN are emitted from the vegetation and are diluted in the ambient air when transported further (above the canopy). Additionally, we found rather flat freezing spectra when aerosol was sampled further away from the sources. This suggests that background aerosol contributes to sampled IN resulting in a heterogeneous mixture. In addition to UA4, on the next day (UA5) IN concentration was rather low in zone 3 and below blank level in zone 2. Furthermore, also zone 0 on UA5 showed lower concentrations compared to UA4. This may be due to the larger time lag between the rain event and sampling or because relative humidity was lower on UA5 compared to UA4. Regarding the freezing temperature, high number concentrations of IN were found to be active around −20 °C, which is similar to the freezing temperatures of silver birch extracts (*Betula pendula*) [3,38] and some extracts from frost resistant vegetation (e.g., black currant (*Ribes*) and sea buckthorn (*Hippophae*)) [53]. In a recent study, we proposed that INMs of *Betula pendula* can be washed off the trees' surface during rainfall events [38]. In more detail, the study investigated the freezing behavior of aqueous extracts from the

surface of birch tissues and compared the results to the rain samples that were collected underneath the birches. Average values of the samples that froze mostly heterogenous (>75%) are compared to the results of this study. Note that the technique of the freezing assays was different (surface and rain samples were measured with the Vienna Optical Droplet Crystallization Analyzer, VODCA [38]). We observed that the average freezing temperature of the three sample types are quite similar (surface: −22.5 °C; rain: −21.6 °C; aerosol: −21.4 °C). However, the data needs to be treated with caution due to different droplet volume. Nevertheless, a comparison of microliter freezing assays and VODCA showed similar freezing results in a previous comparison study [54].

The size of IN in the impinger sampled in the alpine environment were mostly between 300 kDa (~9 nm considering the IN to be spherical [55]) and 100 nm indicating IN to be in the submicron size range (see Figure 4). Furthermore, we also detected IN between 100 kDa (~6 nm in diameter considering the IN to be spherical [55]) and 300 kDa indicating that these IN are macromolecules [34]. Nevertheless, it is challenging to determine the size of airborne IN and we cannot exclude that e.g., IN in the macromolecular size range were attached to larger particles in the air prior to sample collection with the impinger. Heating the most ice nucleation active sample strongly diminished its INA. Biological INMs are often referred to as ice nucleating proteins [31,44,56,57]. Heating such INMs as high as 98 °C unfolds and disrupts proteins (i.e., diminishes tertiary and quaternary structures) resulting in the loss of its ability to nucleate ice [31]. In addition to biological matter, also certain inorganic IN have been tested in their heat stability, e.g., Zolles, et al. [58] treated different minerals with 250 °C and showed that INA of, e.g., Arizona test dust, decreased only slightly (1 °C difference). Thus, in most studies, the heat treatment effect is related to biological IN rather than mineral dust [59,60]. Hence, the remaining shoulder in the cumulative spectrum at around −24 °C (Figure 4) is considered to correspond to heat stable mineral dust, possibly derived from the surrounding mountains due to abrasion or whirled up from the ground (due to the landscape morphology, long range transport can be excluded). Thus, we propose that INMs from *Betula pendula* surfaces could have contributed to the airborne IN found in the vicinity of birches after rainfall. We hypothesize that INMs could have been extracted during rainfall as already shown by Seifried, et al. [38]. These INMs would have been soluble in water and can be aerosolized during splashing events, as shown in a schematic drawing (Figure 9). Thus, bioparticle emission by rain splash (as described by Joung, et al. [20]) from soil or directly from the surface of birches could contribute to airborne IN.

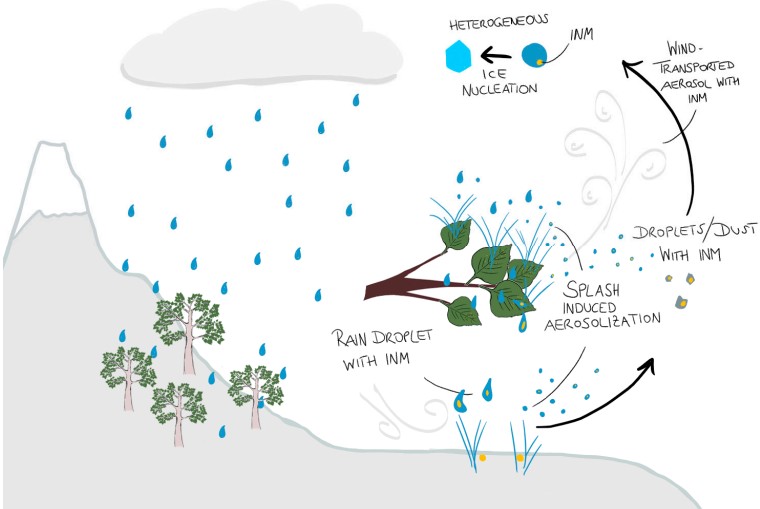

**Figure 9.** Proposed hypothesis of possible aerosolization and transport of INMs from the surface of birches.

Moreover, also different fungal spores are known to be active at −20 °C. For example, rust fungi (e.g., *Endocronartium harknessii*) are pathogens of several pine species and produce ice nucleation active spores (freezing onset temperature, −18 °C) [61]. Beside PBAPs, also inorganic particles, e.g., mineral dust can trigger freezing below −20 °C. In addition, IN can be aerosolized from aqueous systems. Benson, et al. [2] investigated water samples from the lake of this study site (Hinterer Gosausee) and found that 23 out of 72 bacterial colonies showed INA. In contrast to the aerosol samples of this study, however, the freezing onset temperature of the samples was ≥−8 °C. The aerosolization of microbes from waters occurs either through bubble bursting or fragmentation mechanisms. For fresh waters, strong winds (about 3–4 m·s$^{-1}$) are necessary to induce the formation and breaking of waves [25]. In this study, however, the average wind speed at 10 m altitude was lower than 1.2 m·s$^{-1}$ each day. Thus, the source of IN is not easily to determine, and different origin for high IN concentrations near alpine vegetation after rain are presumed.

A more detailed view on the aerosol morphology was obtained from the microscopic analysis of the impactor samples. Thereby, the mean diameter of the sampled aerosol particles in the accumulation and coarse modes on UA4 were 0.17 and 4.0 μm, respectively. The concentration of the particles in the accumulation mode was three orders of magnitude higher than the coarse mode. In both modes, the majority of particles showed auto-fluorescence, indicating a biological origin of the particles. For the accumulation mode, SEM images revealed that the ultra-fine particles tended to agglomerate. Note that agglomeration likely occurred while particles impacted onto the substrate since the agglomerates are larger than the cut-off diameter. Such ultra-fine particles were also found in previous sampling periods without rain events [39], and thus are not linked to rainfall induced aerosols and could represent background sources. Secondary organic aerosol or mineral dust can contribute to accumulation mode aerosol; however, both are known to nucleate ice at lower temperatures as the most abundant IN detected in freezing experiments [62–64]. The morphology and shape of coarse mode particles strongly indicated the presence of cellular material. High fluorescence intensity on the cell-walls of the observed particles supports this assumption [65,66]. Regarding the shape of the cells, about half of the particles appeared spherical, and many fungi of the phylum Basidiomycota have been reported to form cylindrical spores, some of which (e.g., *Agaricus bisporus*, *Amanita muscaria*, *Boletus zelleri*) are known to induce heterogenous ice nucleation [67]. In addition, plants of the alpine ecosystem host several rust fungi e.g., *Endocronartium harknessii*, which produces elliptical spores nucleating ice with a freezing onset temperature of −18 °C [61]. These fungi can be found on mountain pines (*Pinus mugo*) as present in the sampling environment. Such fungal spores can survive long-range transport and therefore could have impact on cloud glaciation and environmental health [68]. Additionally, several bacteria species are known to release spherical spores or appear spherical themselves, but bacteria typically trigger freezing at rather high sub-zero temperatures (>−10 °C) [69,70]. Further, some particles appeared to be coated, likely with an organic film. Huffman, et al. [71] described that organic coating on fungal spores sampled in the Amazonas is of secondary origin. Such film could also originate from extractable IN of plants (e.g., INMs from birch surfaces [3,38], lignin [72], polysaccharides [73] or other biopolymers) and thus, explain the sharp freezing temperature and high IN number concentration found in the ice nucleation experiments.

## 5. Conclusions

In this exemplification study, we investigated atmospheric IN and bioaerosol emissions in an alpine environment with ground and drone-based sampling techniques. Ice nucleation experiments reveal that the dominant IN trigger freezing at about −20 °C, which is associated to IN from birches (immersion freezing mode), with decreasing concentrations towards higher altitudes (ground—canopy—45 m). The characterization indicates IN to be rather heat-sensitive and in the nanometer size range. From the most ice nucleation active sample, about 80% of coarse mode aerosol particles were auto-fluorescent, most of which were biological cells (fungal spores, bacteria). In addition to cellular characteristics (orna-

mented surfaces, defined shapes, etc.), fluorescent coating was observed on certain particles. We hypothesize that part of the sampled IN originate from splash-induced emission of soluble INMs from the birches present at the sampling environment. To further evaluate the atmospheric impact, the transport and atmospheric abundance of IN emitted from the alpine ecosystem has to be investigated in more detail in a long-term study. Extending the sampling intervals and sampling altitude would help to address the atmospheric relevance of IN from birches. In conclusion, the results of our paper and our previous study [38] suggest that vegetation dominated by birches is a potential source of IN extracted and aerosolized after rainfall.

**Author Contributions:** Conceptualization, T.M.S., P.B. and H.G.; methodology, T.M.S., P.B. and A.T.K.; validation, T.M.S. and P.B.; formal analysis, T.M.S., P.B., A.T.K., K.W.; investigation, T.M.S. and P.B.; resources, D.G.S.III, J.F.-N. and H.G.; data curation, T.M.S.; writing—original draft preparation, T.M.S.; writing—review and editing, P.B., A.T.K., D.G.S.III, K.W., J.F.-N. and H.G.; visualization, T.M.S.; supervision, H.G.; project administration, T.M.S.; funding acquisition, H.G. All authors have read and agreed to the published version of the manuscript.

**Funding:** This research was funded by the Austrian Science Fund (FWF), grant number P 26040.

**Institutional Review Board Statement:** Not applicable.

**Informed Consent Statement:** Not applicable.

**Data Availability Statement:** All data are available from the corresponding author upon request.

**Acknowledgments:** The authors would like to thank Nadine Bothen for her technical support with the ice nucleation measurements. T.M.S. and P.B. would like to thank Martin Heinz Stürmer and the Austrian Federal Forests AG for the support and permission to sample in Gosau.

**Conflicts of Interest:** The authors declare that they have no conflict of interest.

**Appendix A**

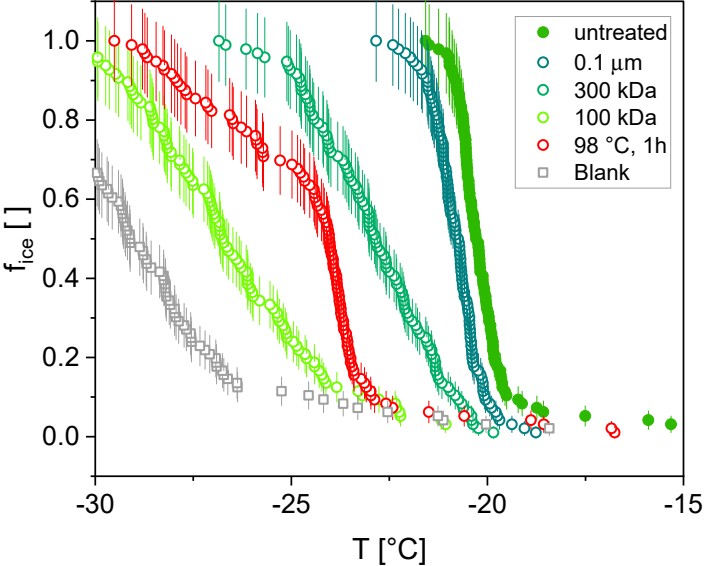

**Figure A1.** Freezing curves of UA4-Imp-Z0: untreated (filled circles, green), filtrated with 0.1 μm cut-off (hollow circles, turquoise blue), 300 kDa cut-off (hollow circles, emerald green), 100 kDa cut-off (hollow circles, light green) and heated, 98 °C for 1 h (hollow circles, red). Milli-Q® water is visualized in grey (blank).

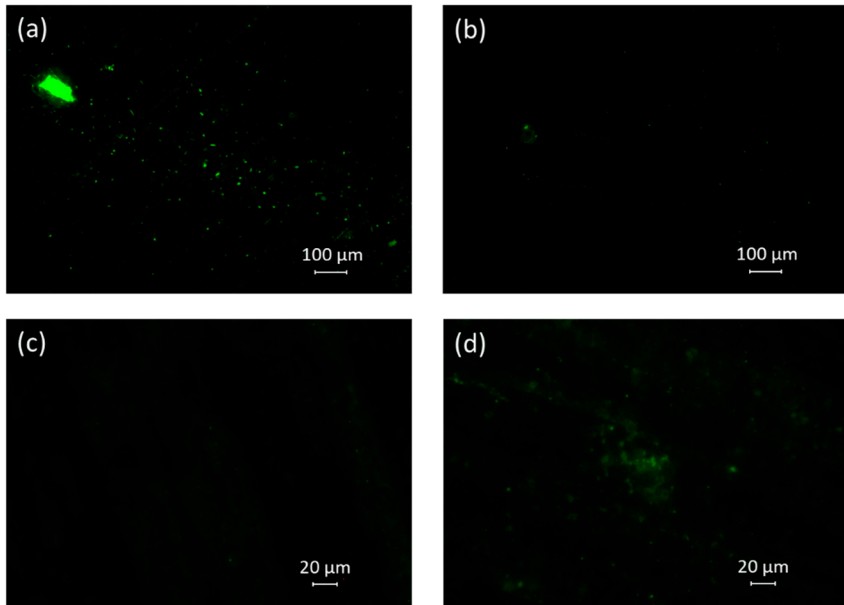

**Figure A2.** UA3-Pac-Z0 fluorescence microscopy pictures: (**a**) stage A, 10× (**b**) stage B, 10×, (**c**) stage C, 40×, and (**d**) stage D, 40×.

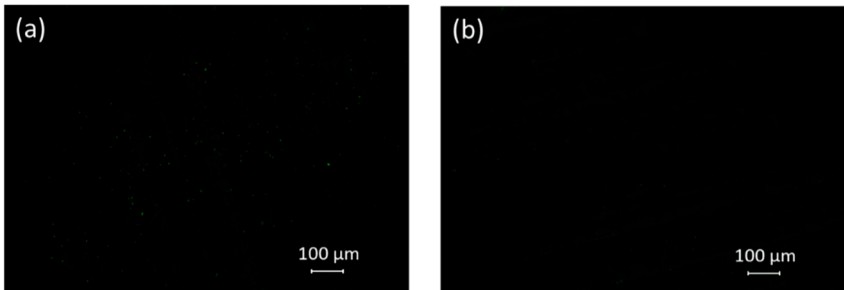

**Figure A3.** UA4-Pac-Z0 fluorescence microscopy pictures: (**a**) stage B, 10× (**b**), and stage C, 10×.

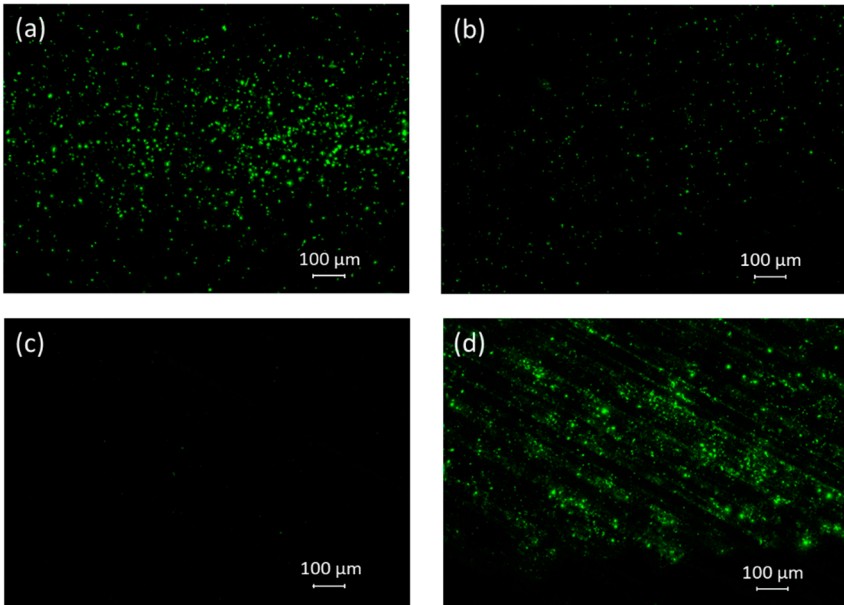

**Figure A4.** UA5-Pac-Z0 fluorescence microscopy pictures: (**a**) stage A, 10×, (**b**) stage B, 10×, (**c**) stage C, 10×, and (**d**) stage D, 10×.

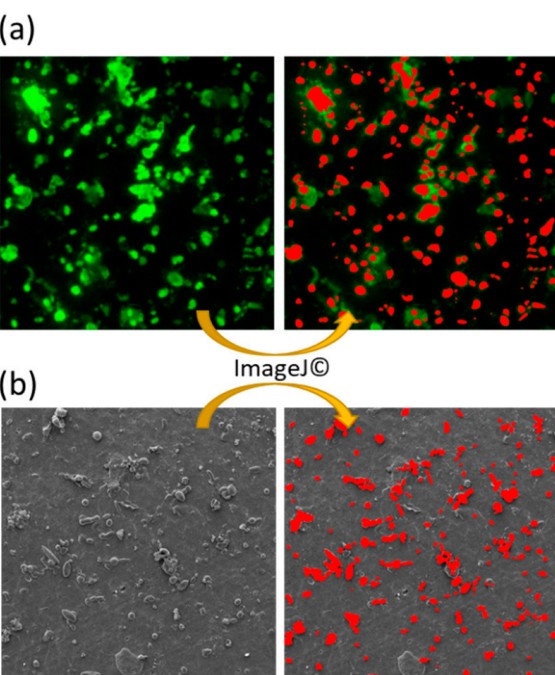

**Figure A5.** Examples of the data evaluation to obtain the number and size distribution of particles impacted on the impactor, stage A. The open software ImageJ was used to count particles in a 200 × 200 μm area. (**a**) shows a fluorescence image and (**b**) a SEM image used for the evaluation. Red areas correspond to analyzed particles.

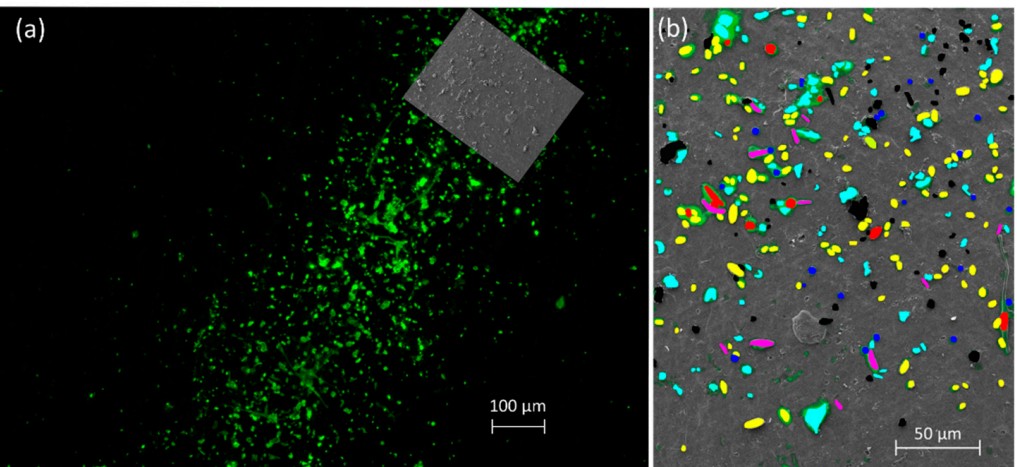

**Figure A6.** (**a**) Overlapped section of fluorescence microscopy and SEM used for particle analysis (UA4-Pac-Z0, stage A). (**b**) Particle analysis (image was rotated 53°): Black coded particles did not show auto-fluorescence properties. The fluorescence active particles were further divided into: yellow corresponds to cylindric, blue to globular, pink to elongate, red to ornamented, and turquois to irregular-shaped particles.

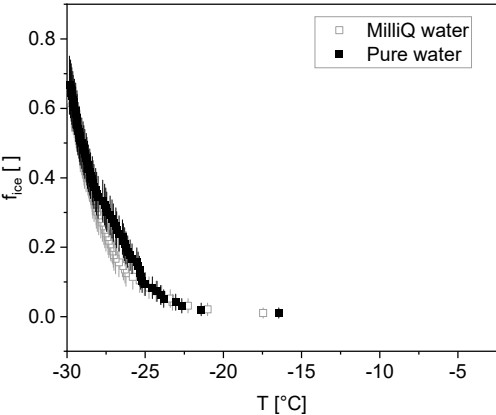

**Figure A7.** Freezing curves of Milli-Q® water (hollow, grey) and pure water (black, filled).

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
