# Peer review of "Ice Nucleation Activity of Alpine Bioaerosol Emitted in Vicinity of a Birch Forest"

_atmosphere, doi:10.3390/atmos12060779_

Round 1

Reviewer 1 Report

This study investigates whether in an alpine environment ice nucleating particles (INPs) can be traced back to birch trees as a source of bioaerosol. To this purpose, samples were collected with an impactor and an impinger at ground level within a birch forest and above its canopy with a rotary wing drone. Ice nucleation was measured with a droplet freezing assay and particles were characterized using fluorescence and electron microscopy.

This is an interesting study that provides convincing arguments that at ground level, there is a prevailing INP type that stems from birch trees and triggers ice nucleation at around -20°C. However, it lacks an adequate discussion that this strong increase in frozen fraction at -20°C is not visible above the forest canopy, suggesting that birch INPs are only one of many INP types present at cloud levels. This finding reveals the need to measure at higher altitudes and shows the aptitude of drones to do this. I suggest that the discussion emphasizes this decrease in birch tree INPs above the tree canopy more explicitly.

This paper is clearly written and generally well presented. However, it has weaknesses and shortcomings regarding the measurement protocol and the evaluation. The 10 min measurement time with the drone above the canopy was too short to collect relevant quantities of INPs. Consequently, there is only a very weak increase of INPs above the blank level for the samples UA5. Longer sampling times could have strongly improved the experimental basis of this study. Moreover, the conversion of the freezing curves to ice nuclei (IN) spectra is erroneous. As is correctly discussed in the text, some freezing curves are not significantly above the blank sample of pure (reference) water. Yet, the corresponding cumulative IN spectra show significant INP numbers. This is because the blank spectrum was not subtracted from the sample spectra. Such a subtraction needs to be performed when the freezing temperature ranges of the pure water reference and the sample overlap. How this should be done can be found in Vali (2019) and David et al. (2019). This manuscript can be considered for publication after these major revisions have been done.

Specific comments:

Line 40: “trigger” is not an adequate verb to characterize a freezing point depression. As a thermodynamic property, a freezing point depression cannot be triggered.

Figure 1: the picture of the sample collection with the drone (panel c) is very small. Consider to enlarge it.

Line 146: can you indicate how far away zone 1 was from the shore of the lake?

Line 172: the evaluation of the freezing curve and the presentation as cumulative IN spectra need to be revised. The blank spectrum needs to be subtracted. This data processing needs to be described in detail (not just by referencing a paper).

Line 173: could you give more information about the water? What water type and what type of filter did you use?

Line 174: was UA4, zone 0, the impinger sample? It would be clearer to use just the abbreviation given in Table 1.

Line 183: What is your pure water sample? The filtered and heated UA4 zone 0 sample?

Line 185: Did you investigate directly the impactor foils? What is the material of the impactor foils?

Lines 249–252: The cumulative spectrum UA3 zone 0 is not evaluated correctly. Therefore, the discussion needs to be adapted after the evaluation has been revised.

Line 256: “The concentration of IN active between…” a word seems to be missing here.

Figure 3: Some of the freezing curves are hardly above the blank sample. When this is the case, the blank IN spectrum needs to be subtracted from the IN spectrum of the sample. See e.g. Vali (2019) or David et al. (2019).

Also, the conversion from freezing curve to IN spectrum of UA4 Z050 (light green) seems erroneous: the freezing curve seems to have a frozen fraction of one at about -21°C while the IN spectrum extends to about -26°C. Yet, it is not possible to obtain an IN spectrum below the temperature of fice = 1 because the freezing curve does not provide any information beyond this range.

Figure 4: This figure needs to be revised. The blank spectrum needs to be subtracted from the sample spectra. It would be best to use pure water references that have undergone the same treatment as the samples since filtration and heating can also influence the IN spectrum of the blank sample. Moreover, it is not clear how the IN spectra shown in Fig. 4 were obtained from the frozen fraction curves shown in Fig. 10. Data displayed in the IN spectra should be limited to the temperature range that the frozen fraction curves cover. Yet the IN spectra continue to temperatures well below a frozen fraction of one. Did you do a dilution series? If yes, this should be reported. Moreover, when the samples were not diluted during filtration or heating, all spectra should cover the same IN concentration range.  

Line 299: what is meant by “considerably large particles”? Specify size.

Figure 5d: Particles seem larger than they should be based on the impactor cutoff. Was this due to agglomeration? Did you measure directly the impactor foils? If yes, how could agglomeration occur? Do you expect such agglomeration also during the measurements in the freezing assay?

Line 347: How did you come to the number of 80 % auto-fluorescent particles? Can you give more details about the evaluation? Looking at Fig. 8a, there seem to be quite many small features that do not fluoresce. On the right half of the image, more particles seem not to fluoresce than to fluoresce. What criterion did you use to classify a feature on the substrate as a particle?

Lines 381–384: You seem to imply here that the particles above the canopy are biological. On what basis do you infer this? Did you measure fluorescence?

Lines 388–390: Indeed, the high number of particles active around -20°C is a strong indication for their origin from birch. Yet, this signature was not present above the canopy, which sheds doubt on the relevance of these particles for cloud glaciation.

Line 401, “volume independent”: what volume do you refer to here? The water volume of the freezing experiment or the air volume?

Lines 405–407: the corresponding size for 100 kDa should also be given.

Lines 416–417: What about the region in the cumulative spectrum at T>-20°C? Some of these INPs resist the heat treatment. This would imply that some of the highest freezing INPs are non-biological.

Line 418: what is meant by “abbreviation”?

Line 424–425: looking at Fig. 3, I come to the contrary conclusion: there are hardly any freezing events at around -20°C observable in zones 2 and 3. Thus, there is a strong dilution of this type of INP above the canopy casting doubt on their relevance for cloud glaciation.

Line 427: the grey data point in panel (a) looks black on my screen. Consider to make it either clearly grey or black.

Line 429, “Note that only samples that froze heterogeneously >75% were taken into consideration”: can you explicitly state which samples you considered? UA3 and UA5?

Line 451: Can you speculate how agglomeration occurred?

Line 456: You state that mineral dust nucleates ice at lower temperatures than observed in the samples that you investigated in this study. Yet, mineral dust also contains nucleation sites that nucleate ice up to -2°C. See e.g. Harrison et al., (2016) and Whale et al. (2017).

Line 469, …but bacteria typically trigger freezing at rather high sub-zero temperatures (>-10°C): Can you give references here?

Figure 11: You do not discuss that UA3 has the highest concentration of INP freezing above -20°C (as shown in Fig. 3) but exhibits hardly any fluorescence. This would point to a non-biological origin of these INPs.

References:

David, R. O., Cascajo-Castresana, M., Brennan, K. P., Rösch, M., Els, N., Werz, J., Weichlinger, V., Boynton, L. S., Bogler, S., Borduas-Dedekind, N., Marcolli, C., and Kanji, Z. A.: Development of the DRoplet Ice Nuclei Counter Zurich (DRINCZ): validation and application to field-collected snow samples, Atmos. Meas. Tech., 12, 6865–6888,  https://doi.org/10.5194/amt12-6865-2019, 2019.

Harrison, A. D., Whale, T. F., Carpenter, M. A., Holden, M. A., Neve, L., O’Sullivan, D., Vergara Temprado, J., and Murray, B. J.: Not all feldspars are equal: a survey of ice nucleating properties across the feldspar group of minerals, Atmos. Chem. Phys., 16, 10927–10940, https://doi.org/10.5194/acp-16-10927-2016, 2016.

Whale, T. F., Holden, M. A., Kulak, A. N., Kim, Y.-Y., Meldrum, F. C., Christenson, H. K., and Murray, B. J.: The role of phase separation and related topography in the exceptional ice-nucleating ability of alkali feldspars, Phys. Chem. Chem. Phys., 19, 31186–31193, https://doi.org/10.1039/C7CP04898J, 2017.

Vali, G.: Revisiting the differential freezing nucleus spectra derived from drop-freezing experiments: methods of calculation, applications, and confidence limits, Atmos. Meas. Tech., 12, 1219–1231, https://doi.org/10.5194/amt-12-1219-2019, 2019.

Reviewer 2 Report

Attached PDF

Round 2

Reviewer 1 Report

The authors have followed my suggestions and revised the manuscript accordingly. It can now be published subject to the following minor corrections:

Line 38: Why in the upper troposphere? In the upper troposphere, temperatures are usually low enough for homogeneous ice nucleation. “Free troposphere” as it was written in the first submitted version would be more appropriate.

Line 47: “correlated with” instead of correlated to”

Figure 1 appears three times in my version of the revised manuscript. Please correct.

Line 455: “size airborne IN”: “of” is missing

Line 457: the point is missing between “INA” and “Biological”.
